# Maternal Melatonin Contributes to Offspring Hair Follicle Development Through Transcriptional Regulation of the AP-1 Complex and MAPK Pathway

**DOI:** 10.3390/ijms26051952

**Published:** 2025-02-24

**Authors:** Yang Feng, Ruixin Yang, Jianqiang Zhang, Haonan Yuan, Zunqiang Yan, Pengfei Wang, Xiaochun Ma, Ting Liu, Shuangbao Gun

**Affiliations:** 1College of Animal Science and Technology, Gansu Agricultural University, Lanzhou 730070, China; 2College of Animal Science and Technology, Shihezi University, Shihezi 832003, China; 3Institute of Animal Science, Chinese Academy of Agricultural Sciences, Beijing 100193, China

**Keywords:** hair follicle, melatonin, AP-1, offspring, Rex rabbit

## Abstract

Maternal melatonin (MT) readily crosses the placental barrier to enter the fetal circulation, and it holds the potential to enhance hair follicle (HF) development, possibly augmented through nutritional interventions during pregnancy. However, the specific impact of maternal MT treatment on fetal HF development remains largely unexplored. In this study, we implanted pregnant rabbits with 10 mg of MT-containing and non-MT-containing silica gel microcapsules. We then assessed HF density and the extent of HF cell apoptosis in the neonatal rabbits. Our findings revealed that maternal MT implantation significantly reduced HF cell apoptosis and promoted an increased HF density in the neonates. Mechanistically, this process involved MT downregulating the expression of JUN/FOS and AP-1, while concurrently upregulating equol expression and reducing norepinephrine levels. Analysis of key protein expression within the MAPK pathway indicated that maternal MT activated this pathway. These results suggest that maternal MT treatment promotes beneficial HF development in offspring. Notably, the transcriptional regulation of JUN/FOS members of the AP-1 complex emerges as a pivotal factor mediating the beneficial effects of MT on neonatal hair follicle development.

## 1. Introduction

As one of the most distinctive characteristics of mammals, hair performs various functions, including thermoregulation, physical protection, and sensory functions. The hair follicle (HF), which is an important skin accessory, produces terminally differentiated keratinocytes that form the hair shaft [1,2]. The HF can also regenerate periodically and is highly modifiable. Most postnatal regulatory processes of HFs are directed toward the regulation of growth phase duration, such as increasing and decreasing the duration of the anagen and catagen phases, respectively [3,4]. However, promoting HF development by changing the durations of the growth phases shows limited efficacy. Therefore, novel improvements in HF density are warranted.

Hair folliculogenesis, a complex developmental process involving precisely regulated ectodermal–mesodermal interactions, is completed during the embryonic period [1]. In mammals, including lagomorphs such as rabbits, the initiation of hair follicle morphogenesis occurs during the pre-embryonic stage, marking the earliest phase of cutaneous appendage development [5]. Mature hair follicles that have completed their developmental program and entered the active growth phase are anatomically characterized by their deep anchorage within the subcutaneous tissue layer. Given that increasing the number of HFs seems to have no role in postnatal animals, it is crucial to investigate the potential impact of increased HF density during the fetal period. Fetal development encompasses a programming process that involves the interaction of genetic information and environmental factors, resulting in the formation of highly adaptable phenotypes [6]. The organism can alter its phenotype in response to environmental changes, and the new phenotype becomes permanent; this is considered a change in programming [7,8]. Thus, improving HF development is crucial during the fetal period; however, to our knowledge, there is limited research on this topic.

Melatonin (MT), also known as *N*-acetyl-5-methoxytryptamine, has various biological activities, such as hormonal, neurotransmitter, immunomodulatory, and biomodulatory effects [9,10]. In mammals, melatonin synthesis is not confined to the pineal gland but extends to various peripheral organs and tissues [11]. Among these, the skin occupies a unique position at the interface between external and internal environments, serving as a critical site for local melatonin synthesis and metabolism. This cutaneous melatoninergic system plays a pivotal role in maintaining skin homeostasis through the regulation of mitochondrial function in epidermal cells, thereby influencing their phenotypic characteristics. The evolutionary conservation of melatonin and its metabolites underscores their fundamental role in cellular protection, particularly in mitigating oxidative stress and restoring cellular homeostasis [12]. Furthermore, the presence of membrane-bound and putative nuclear melatonin receptors in skin cells provides a molecular basis for melatonin’s regulatory effects on cutaneous physiology and structural integrity [13]. The pilosebaceous unit, consisting of hair follicles and their associated sebaceous glands, represents one of the most functionally significant cutaneous appendages. MT has been demonstrated to play a pivotal regulatory role in multiple aspects of pilosebaceous physiology, including the modulation of hair growth cycles, regulation of pigmentation processes, and control of melanoma progression [14]. Experimental evidence from fur-bearing animal models has established that MT exerts significant stimulatory effects on follicular growth [15,16], while clinical studies have documented its capacity to extend the anagen phase duration in female patients with androgen etic alopecia [17]. Mammalian embryonic development mostly depends on maternal MT supplied through the placenta, which crosses the placental barrier without modification [18]. In humans, the functional significance of this maternal-fetal MT transfer is further underscored by the presence of MT receptors in the suprachiasmatic nucleus (SCN) of both adult and fetal brains, enabling direct physiological effects on fetal development [19]. The physiological importance of MT is further evidenced by its ability to cross all biological barriers without metabolic alteration, including the placental interface [19], and its demonstrated involvement in placental function across mammalian species, including humans [20]. Rodent studies have provided additional mechanistic insights, identifying specific MT binding sites in fetal pituitary tissue and demonstrating a progressive elevation of maternal MT concentrations throughout gestation, with peak levels observed at parturition [21,22]. This temporal pattern of MT availability coincides with critical stages of embryonic and fetal development, highlighting the essential role of maternal MT supply prior to the maturation of the fetal pineal gland, which occurs postnatally. Mammals are sensitive to hormone concentrations during embryonic development, and disturbances in the synthesis and secretion of maternal hormones can affect the development of vital organs and tissues in the offspring [23]. Maternal supplementation of MT protects against programmed cardiovascular disease and is a potential therapeutic target for clinical interventions against developmental disorders [24]. Alterations to the in utero environment due to maternal hormone changes may have profound and long-term implications for the developing fetus through epigenetic mechanisms. However, research on the effect of maternal MT on fetal HF development is limited. Our previous study was the first to prove that MT promotes HF density and decreases hair diameter [25]. However, the effect of maternal MT remains unclear for several other key parameters, such as HF cell apoptosis, and the underlying mechanisms need to be elucidated. Additionally, this topic has remained largely understudied. The folliculogenesis process in mammals exhibits similarities, and the Rex rabbit, with its high density of hair follicles, constitutes an ideal animal model for the study of hair follicles.

This study aimed to determine whether maternal MT treatment during pregnancy benefits the developmental programming of HFs and investigate the mechanisms underlying the function of maternal MT in offspring HF development. Accordingly, we identified differentially expressed genes (DEGs) and metabolites in the skin of the various treatment groups. Our study provides a valuable basis for ascertaining the role of maternal MT in gene expression and metabolic responses in the development of offspring HFs, thus providing valuable insights into the causes of hair disorders.

## 2. Results

### 2.1. Maternal MT Supplementation Decreases HF Cell Apoptosis and Promotes the HF Density in Neonatal Rabbits

The TUNEL assay was used to detect apoptosis in the HF cells of rabbit neonates. The MT group showed fewer TUNEL-positive apoptotic cells and limited DAPI staining compared to that in the CR group (Figure 1a). The apoptosis rate in the MT group was significantly lower than that in the CR group (*p* < 0.01, Figure 1b). This implies that maternal MT reduces HF cell apoptosis in offspring. In addition, as reported in our previous study, the MT group showed higher and lower degrees of hair density and hairiness, respectively (Figure 1c,d, Appendix A) [25].

### 2.2. FOS and JUN Play a Crucial Role in the Maternal Transmission of MT to Offspring

In total, 17,296 mRNAs were expressed in the two groups; 2532 DEGs were detected, of which 1131 were upregulated and 1401 were downregulated (Figure 2a). qRT-PCR was performed to validate the RNA-Seq results, and the fold changes in the expression levels of the 21 key genes were found to be consistent between the qRT-PCR and RNA-seq data (Figure 2b). We used 12 algorithms to screen the hub genes (the top 30 genes and scores of each algorithm are shown in Appendix A). Through the gene score and function (enriched GO terms of DEGs are shown in Appendix A), a correlation analysis of hub gene expression with the HF developmental phenotype and HF apoptosis (Figure 2c), and an interaction and network analysis (Figure 2e), we found that the key genes *JUN* and *FOS* play crucial roles on offspring HF development after maternal MT implantation. 

The mRNA expression levels of FOS and JUN, as well as the protein expression of FOS and JUN in the skin, were significantly downregulated in the MT group (Figure 3a). The IHC findings further indicated a downward trend in the expression of FOS and JUN in the MT group (Figure 3b,c). Additionally, the serum AP-1 levels were significantly lower in the MT group compared to those in the CR group (Figure 3e).

### 2.3. Maternally Transmitted MT Affects the Development of HF in Offspring by Regulating the MAPK Signaling Pathway

KEGG pathway enrichment was performed to elucidate the biological functions of the DEGs (Figure 2d). The MAPK pathway was significantly enriched (13 upregulated genes and 46 downregulated genes) and associated with HF development. *JUN* and *FOS* were both enriched in the MAPK signaling pathway. MAPK was activated in the process of MT transmit. We measured the key protein in the pathway, and the results showed that *FGF18*, and *NGF* were significantly downregulated, while *PDGFRB* and AXIN2 were significantly upregulated (Figure 3d). The protein expression levels of FOS, JUN, NGF, FGF18, AXIN2, and PDGFRB were consistent with their mRNA expression levels (Figure 2b and Figure 3d). 

### 2.4. Downstream Metabolites of FOS/JUN, Norepinephrine, and Equol Were Significantly Altered in the Neonatal HF Development of the MT Group 

The comprehensive metabolic profiling results are systematically presented in Figure 4, Appendix A. (Specifically, Appendix A illustrates the DSPC interaction network of the identified metabolites. Appendix A displays the top 25 most significantly enriched KEGG pathways based on metabolic pathway analysis. Furthermore, Appendix A provides the correlation analysis of KEGG pathways between differential genes and metabolites in positive ion modes. Appendix A shows the differential metabolites between the MT and CR group) The network analysis of DEGs and DEMs showed that *FOS* and *JUN* were located at the key nodes, suggesting their crucial roles in the process, as our preliminary results had indicated. The signature metabolites norepinephrine and equol were also included. Equol emerged as a pivotal node in the interaction map downstream of the *JUN*/*FOS* genes, while norepinephrine occurred downstream of *FOS* (Figure 4e). Both norepinephrine and equol exhibited area under the curve (AUC) values ranging from 0.8 to 1, and their expression levels significantly differed between the two groups (Figure 4f). These findings confirm their roles as signature metabolites in the process.

## 3. Discussion

### 3.1. Maternal Implantation of MT Improved Fetal HF Developmental Programming

Maternal nutrition plays an important role in fetal growth. Nutritional programming is essential for offspring, and it can have long-lasting or even lifelong effects. Some reports have shown that MT reprogramming during pregnancy can impact various biological processes in the offspring over the long term, including oxidative stress [26,27], epigenetic regulation [28], reduction in the number of nephrons [29], and mutations in *RAS* [30]. Our previous investigations have demonstrated significantly elevated serum MT concentrations in both neonatal and maternal rabbits from an MT-treated group compared to controls (*p* < 0.05) (Appendix A), accompanied by altered prolactin (PRL) levels during the neonatal period (*p* < 0.05) (Appendix A). These findings provide compelling evidence for the successful transplacental transfer and biological activity of maternal MT supplementation. Although the skin is well-established as an active site of local melatonin synthesis in adult organisms, where it exerts regulatory control over mitochondrial function and modulates cellular phenotypic characteristics [12], the potential for melatonin synthesis and metabolic activity in fetal skin remains to be fully characterized. Despite this uncertainty regarding fetal cutaneous melatonin metabolism, our experimental data clearly demonstrate that maternal MT supplementation significantly enhances follicular density, affecting both SF and PF populations in offspring [25]. The current study further demonstrates that maternal MT treatment reduces HF cell apoptosis in developing offspring. Supported by neonatal serum MT level measurements [25], these findings confirm that maternally administered MT effectively crosses the placental barrier, enters fetal circulation, and influences HF development. This transgenerational effect of maternal MT supplementation highlights its potential role in regulating fetal cutaneous development. Apoptosis is not only a type of cell death but also an essential biological process regulated by complex molecular mechanisms. Immature SFs had a decreased density during maturation due to apoptosis [31], which could explain why the SF density in the CR group was lower than that in the MT group in this study. MT exerts powerful antioxidant properties to protect cells by directly scavenging free radicals and stimulating antioxidant enzyme activity [32,33,34]. In previous studies, it has been suggested that MT holds the potential to decrease HF cell apoptosis. Specifically, the implantation of MT in cashmere goats was observed to enhance the activity of antioxidant enzymes, indicating its protective role against oxidative stress [32]. Similarly, the administration of MT during pregnancy was found to reduce apoptosis in neonatal rats, further supporting its anti-apoptotic effects [35]. Building upon these findings, our current study demonstrates that MT not only exhibits anti-apoptotic properties but also has the ability to reprogram HF development.

### 3.2. Transcriptional Activation of JUN/FOS Members of the AP-1 Complex Is a Key Factor in the Process of the MT Effect on the Neonate Hair Follicles

*FOS* is a member of the FOS protein family, which encode leucine zipper proteins that bind to proteins of the JUN family [36,37]. FOS plays a pivotal role in regulating cell proliferation, differentiation, transformation, and apoptosis. JUN, alternatively known as cJUN, is a member of the JUN family of proteins and has the ability to homodimerize with FOS family proteins, forming the dimer activator protein-1 (AP-1). This protein complex is involved in various cellular processes, including cell proliferation, differentiation, and apoptosis [37]. Evidence suggests that the AP-1 transcription factor plays a critical role in the development and growth of mammalian embryos [38,39]. Some studies have reported specific expression patterns (timing and target organs) of this molecule during embryogenesis [40]. However, there have been limited reports exploring the impact of maternal melatonin (MT) on skin-related issues. An intriguing observation is that MT promotes the development of secondary hair follicles in adult cashmere goats by inhibiting AP-1 [41]. In our study, we found that both JUN and FOS expression levels were significantly downregulated in the MT-treated group. This was further confirmed by Western blotting and immunohistochemistry results, which unequivocally demonstrated a marked downregulation of FOS/JUN proteins. Additionally, a notable decrease in serum AP-1 levels was observed in the neonatal offspring belonging to the maternal MT group. Some studies have shown that AP-1 could promote apoptosis [42,43], suggesting that the downregulation of AP-1 may be a key factor in the mechanism by which maternal MT affects offspring hair follicle development. This finding provides a plausible explanation for the reduced apoptosis rate observed in the MT group in comparison to the control group.

During our investigation, we observed that the mitogen-activated protein kinase (MAPK) pathway, which is a type of intracellular serine/threonine protein kinase and an important component of the intracellular signaling system, was activated. This pathway is associated with HF distribution, vellus hair growth, and overall HF growth [44,45,46]. Notably, in our study, both JUN and FOS were found to be enriched in the MAPK pathway, which in turn promoted HF development.

### 3.3. Downstream Metabolites of FOS/JUN, Norepinephrine, and Equol Seem to Play a Key Role in the Neonatal Hair Follicle Development of the MT Group

The skin, a complex neuroendocrine organ, serves as a target for various hormones, neurohormones, neurotransmitters, and neural responses. Changes in the hormonal milieu have been shown to significantly impact HF development [47]. The present study suggests that maternal MT upregulates equol expression and downregulates norepinephrine levels during this process. Equol, an anti-androgenic molecule that inhibits prostate growth and hormonal feedback, is an important factor that regulates reproductive function [48]. In our study, equol was identified as a DEM. A large proportion of mammalian hair loss is associated with androgens, and it has been shown that equol inhibits 5α-reductase activity, prevents the conversion of testosterone to dihydrotestosterone, stimulates energy production, and reduces the incidence of hair loss [49]. In contrast, norepinephrine operates within a negative regulatory feedback loop, inhibiting its own secretion while inducing the secretion of MT [50]. Intriguingly, a previous study reported that the number of nerve fibers containing norepinephrine or immunoreactive for tyrosine hydroxylase increases during the early anagen phase of the hair cycle but decreases during the late anagen phase [51]. This dynamic regulation of norepinephrine suggests its involvement in the HF developmental process. Notably, both norepinephrine and equol appear to be downstream metabolites of FOS/JUN. The alterations in their expression suggest that maternal implantation of MT influences hair follicle development through neurogenic factors, potentially via the activation of pathways involving the FOS/JUN complex. This mechanism provides a plausible explanation for how maternal MT promotes offspring HF development by modulating the expression of these two metabolites. 

From another perspective, peroxisome proliferator-activated receptor gamma (PPARγ) has been identified as a potential therapeutic target for melatonin and its bioactive metabolites [52]. PPARγ is a nuclear receptor abundantly expressed in cutaneous tissues, and PPARγ-mediated signaling pathways have been demonstrated to modulate mitochondrial energy metabolism within human hair follicle epithelial cells [53]. Mechanistically, melatonin and its indoleamine and kynurenine metabolites exhibit dual regulatory properties, functioning as aryl hydrocarbon receptor (AhR) agonists while simultaneously interacting with PPARγ at elevated concentrations [52]. Furthermore, experimental evidence suggests that equol, a biologically active isoflavone metabolite, enhances PPARγ transcriptional activity, whereas norepinephrine exerts inhibitory effects on PPARγ gene expression [54,55]. In the current investigation, the observed upregulation of equol concomitant with downregulation of norepinephrine suggests a potential mechanistic pathway for modulating PPARγ activity. 

Notably, melatonin is naturally present in various nutritional products and can be effectively absorbed through the gastrointestinal (GI) tract. This raises the possibility that a portion of fetal MT levels may originate from GI-mediated absorption and subsequent metabolic processing, suggesting an alternative pathway for MT delivery during fetal development. However, this potential mechanism requires further investigation to determine its relative contribution to overall fetal MT metabolism [56]. 

In conclusion, our study has revealed that maternal MT administration significantly enhances HF density and reduces HF cell apoptosis in neonatal offspring. As illustrated in Figure 5, after implantation during pregnancy, MT crosses the placental barrier and acts on fetal skin. During this process, MT downregulates the expression of JUN/FOS and AP-1, while simultaneously upregulating equol expression and downregulating norepinephrine levels. These alterations are apparently mediated through the stimulation of the mitogen-activated protein kinase (MAPK) pathway regulation. Our experimental data suggest that maternal MT administration might influence HF development in offspring, though further investigation is required to establish this relationship conclusively. These preliminary findings indicate a potential direction for exploring therapeutic strategies targeting MT signaling pathways in HF-related conditions, pending additional validation through more comprehensive studies.

## 4. Materials and Methods

### 4.1. Animal Management and Experimental Design

This animal study complied with the 1988 Regulations for the Administration of Affairs Concerning Experimental Animals (State Science and Technology Commission of the People’s Republic of China) and was approved by the Gansu Agricultural University Animal Welfare and Ethical Review Board (Approval Code: 20122159, Approval Date: 15 June 2018). We artificially inseminated (AI) 93 parous female rabbits with frozen sperm sourced from the same male rabbit. Pregnancy was determined on the 10th day after AI, and 79 pregnant rabbits were included in this study. The pregnant rabbits were randomly divided into two groups, the MT treatment (MT group, n = 40) and control groups (CR group, n = 39). In the MT group, MT microcapsules (containing 10 mg of MT and silica gel; provided by the Institute of Special Animal and Plant Sciences, Changchun, China) were implanted subcutaneously into the rabbits. Microcapsules of the same size containing only silica gel were implanted into the rabbits of the CR group. All microcapsules were implanted in the dorsum of the neck. All implantations were performed gently to prevent abortions. All rabbits were raised in natural light with food and water provided ad libitum. The diet (detailed in Appendix A) was prepared in accordance with the dietary specifications recommended by the United States National Research Council.

### 4.2. Skin Tissue Collection, Processing, Terminal Deoxynucleotidyl Transferase-Mediated dUTP Nick-End Labeling (TUNEL) Assay, and Immunohistochemical (IHC) Study

At birth, seven neonates from various litters of the two groups were euthanized via cervical dislocation and immediately skinned. Next, 1 cm × 1 cm skin samples from the center of the back were collected for RNA-seq and metabolic profiling analysis. The skin samples were then frozen using liquid nitrogen and stored at −80 °C before RNA extraction. Another set of skin samples (1 cm diameter) from the back were collected and immediately soaked in a paraformaldehyde fixative solution after being rinsed with normal saline. The TUNEL assay is detailed in the Appendix A. In the IHC study, the following primary antibodies were utilized: anti-FOS (sourced from Proteintech, Wuhan, China; catalogue number: 66590), a rabbit monoclonal primary antibody, clone M1; and anti-JUN (also sourced from Proteintech, Wuhan, China; catalogue number: 66210), another rabbit monoclonal primary antibody, clone M1. The IHC staining process was automated using the Ventana BenchMark Ultra instrument (V 5.0, Roche Diagnostics, Tucson, AZ, USA). For the quantification of the target protein expression levels, Image-Pro Plus version 6.0 software (Media Cybernetics, Rockville, MD, USA) was employed.

### 4.3. RNA-Seq Analysis and Identification of Key Genes

We analyzed 14 samples (n = 7 each for MT and CR groups) by RNA-seq (Appendix A, sequence of the primers was shown in Appendix A). The DEGs were loaded into an interaction network, and the key node genes were determined using a network topology algorithm (Appendix A). The correlation between gene expression and HF development was analyzed through Spearman’s correlation analysis. A protein–protein interaction (PPI) network was constructed using the STRING database (https://string-db.org/) (accessed on 1 June 2024) to further investigate the interactions between the DEGs. By hiding disconnected nodes with confidence >0.9, a visual representation of the PPI network was obtained using Cytoscape software (version 3.6.1).

### 4.4. Quantitative Real-Time Reverse Transcription-Polymerase Chain Reaction (qRT-PCR) and Western Blotting

The expression levels of 21 genes were measured, including 13 key genes screened during RNA-seq and 8 genes related to HF development. Quantification of the protein expression of FOS, JUN, NGF, FGF18, AXIN2, and PDGFRB was performed as previously described [57]. Details of the qRT-PCR and Western blot analysis are given in the Appendix A.

### 4.5. Serum Collection and the Measured AP-1

Blood samples from neonatal rabbits were collected in tubes without anticoagulant and centrifuged at 3000 revolutions per minute for 10 min. The serum was then isolated and analyzed for AP-1 levels using ELISA kits (provided by Westang Biotech Co., Shanghai, China).

### 4.6. Metabolic Profiling 

The methods for metabolic profiling and analysis are provided in the Appendix A.

### 4.7. Correlation Analysis Between DEGs and DEMs

The Pearson correlation coefficient was used to analyze the degree of association between DEGs and DEMs, where correlation coefficients > 0 and < 0 indicate a positive and negative correlation, respectively. The main biochemical and signal transduction pathways associated with the DEGs and DEMs were identified using the Kyoto Encyclopedia of Genes and Genomes (KEGG) database. The ggplot2 package in R (version 3.2.1) was used to determine the pathways wherein metabolism and transcription were co-enriched, based on the corresponding bubble maps. The interaction network of the key DEGs and DEMs was analyzed using the MetaboAnalyst Online Analysis System and visualized using Cytoscape software (version 3.6.1).

### 4.8. Statistical Analysis

SPSS software (version 20.0) was used to perform statistical analyses regarding HF development. An unpaired Student’s *t*-test was used to determine the significance of differences between the MT and CR groups, with significance set at *p* < 0.05.

## Figures and Tables

**Figure 1 ijms-26-01952-f001:**
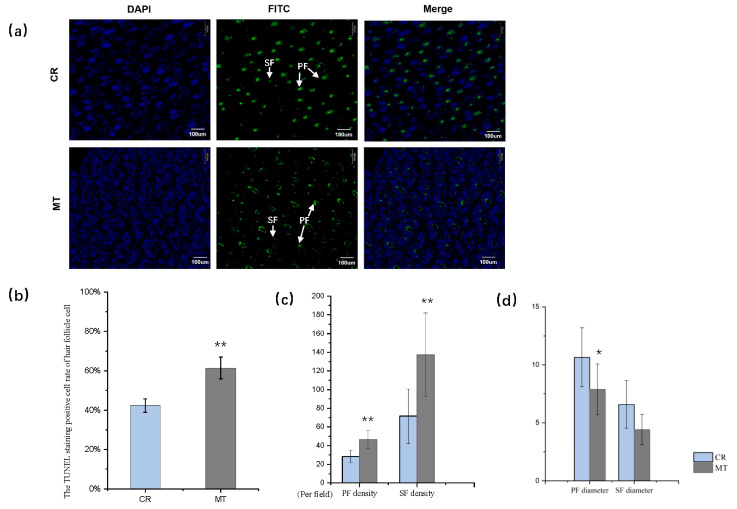
(**a**) Terminal deoxynucleotidyl transferase-mediated dUTP-nick end labeling TUNEL staining of rabbit neonate skin samples in the MT and control (CR) groups. TUNEL staining (green, middle panel) indicates apoptosis-positive cells, DAPI staining (blue, right panel) indicates nucleated cells, and the merged column (left panel) shows cells stained with both TUNEL and DAPI, White arrows: Indicate the primary follicles (PF) and secondary follicles (SF). (**b**) Comparison of the rates of TUNEL-positive staining in the hair follicle cells of the two groups. (**c**) Densities of primary and secondary follicles of neonatal rabbits in the two groups. (**d**) Hair diameters of primary and secondary follicles of neonatal rabbits in the two groups. * indicates *p* < 0.05, ** indicates *p* < 0.01.

**Figure 2 ijms-26-01952-f002:**
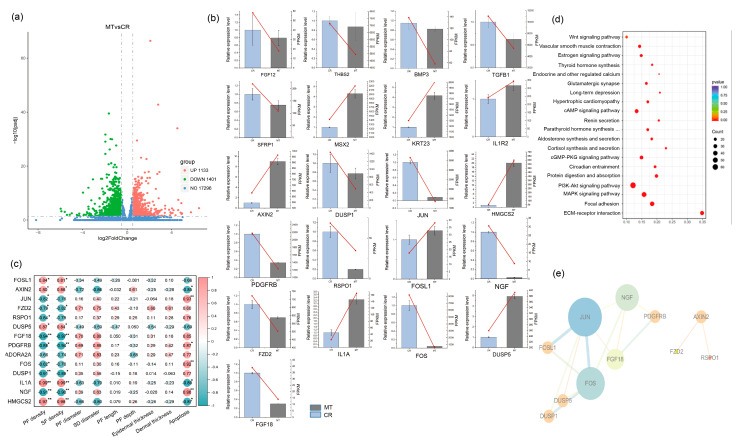
(**a**) Volcano map of the differentially expressed genes (DEGs). (**b**) RT-qPCR validation of the RNA-Seq results. The bar graph shows the qPCR results, while the red line graph shows the RNA-seq values. (**c**) Heat map of the key genes correlated with HF development and HF cell apoptosis. (**d**) The significantly enriched Kyoto Encyclopedia of Genes and Genomes (KEGG) pathways of the DEGs. The top 20 pathways with the highest enrichment scores are shown in descending order. The red dots indicate significant differences in gene expression levels in the corresponding signaling pathway compared to the control group. (**e**) The interaction network and expression of key genes. The sizes of the circles indicate the degree values, and the thickness of each line indicates the relationship degree between two genes (combined score). * indicates *p* < 0.05, ** indicates *p* < 0.01.

**Figure 3 ijms-26-01952-f003:**
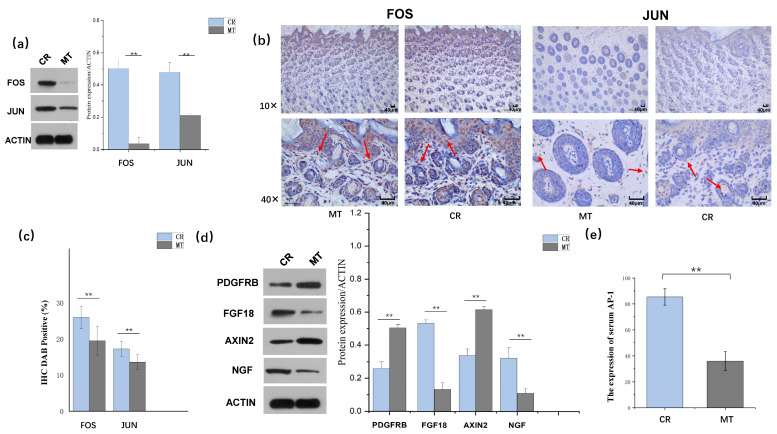
(**a**) Western blot analysis of FOS/JUN differentially expressed proteins. ** indicates *p* < 0.01. (**b**) Immunohistochemical analysis of FOS (left) and JUN (right) in the MT and CR groups. The red arrow points to the expression of FOS and JUN proteins in MT group and CR group. (**c**) DAB positivity in immunohistochemistry in the MT and CR groups. (**d**) Western blot analysis of key differentially expressed proteins. (**e**) Melatonin regulates the expression levels of serum AP-1 in the offspring rabbit.

**Figure 4 ijms-26-01952-f004:**
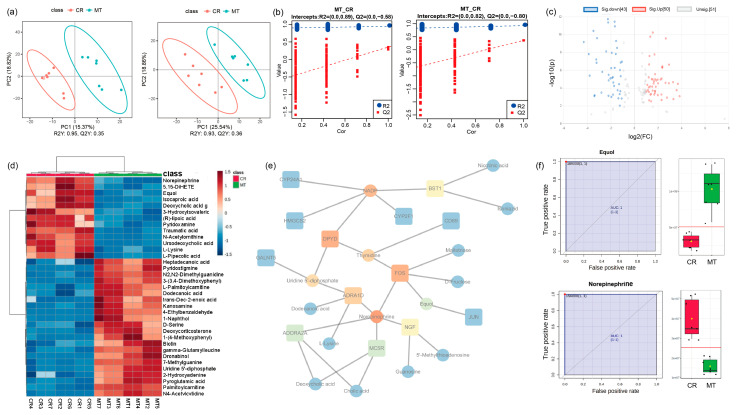
(**a**) OPLS-DA plots in the positive ion and negative ion modes. The left side shows data from the positive ion mode, while the right side shows data in the negative ion mode. (**b**) Permutation test of the OPLS-DA model in the positive and negative ion modes. (**c**) Volcano plot of the differential metabolites in the skin of the MT vs. CR groups. (**d**) Heat map of the differential metabolites in the skin of the MT vs. CR groups. We conducted hierarchical clustering separation of individual sample points for the DEMs, and we present the distances between the individual sample points for the 35 most significant DEMs in each group in a heat map. (**e**) The network of differential genes and differential metabolites. The square nodes are genes, while the circular nodes are metabolites. FOS interacts with thymine, maltotriose, norepinephrine, D-fructose, and equol, while JUN interacts with equol. Among the metabolites, equol and norepinephrine are located at key nodes. (**f**) The ROC work curve of equol and norepinephrine and the expression in the MT and CR groups. Black dots represent specific data points on the ROC curve.

**Figure 5 ijms-26-01952-f005:**
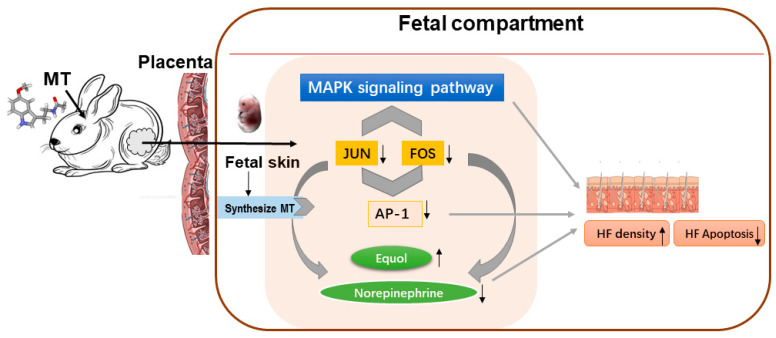
Mechanism underlying the effective interventions of maternal melatonin on the hair follicles of rabbit offspring. The direction of the black arrow indicates the up-regulation and down-regulation of the corresponding gene.

## Data Availability

Data are openly available in a public repository. The transcriptome data that support the findings of this study are openly available in NCBI at https://dataview.ncbi.nlm.nih.gov/object/PRJNA883925?reviewer=a02btju0l98a6slickq4meskec (accessed on 20 June 2024). Metabolomic data are openly available in MetaboLights under MTBLS6009 “Maternal melatonin: Effective interventions in the beneficial developmental programming of hair follicles in offspring” (https://www.ebi.ac.uk/, accessed on 20 June 2024).

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
