# Peer review of "Maternal Melatonin Contributes to Offspring Hair Follicle Development Through Transcriptional Regulation of the AP-1 Complex and MAPK Pathway"

_ijms, 2025, doi:10.3390/ijms26051952_

Round 1

Reviewer 1 Report

Comments and Suggestions for Authors

This study showed that maternal treatment with MT promotes developmental HF in direct offspring, the main mechanisms involved are: the transcriptional regulation of JUN/FOS members of the AP-1 complex emerges as a pivotal factor in mediating the beneficial effects of MT on neonatal hair follicle development. The research is relatively innovative. However, there are still some key questions need to be resolved:

1. What evidence suggests that MT has crossed the placental barrier? For example, measuring MT in the maternal placenta or fetal blood, or analyzing changes in the maternal uterus internal environments.

2. MT has a significant impact on the reproductive performance of female rabbits. In this study, Does MT have negative effects such as affecting litter size or live birth? No data was found in the study.

3. In the article, relevant genes were identified through skin transcriptome screening, and key genes were tested through RNA and protein analysis. Metabolic profiling was performed on downstream pathways. But they all used relatively single methods, especially in terms of mechanisms, without adding other methods to support them, which feels a bit weak and insufficient for the following conclusions.

4. In Methods, At the same day? Sperm of one male rabbit could AI 93 female? This is obviously unrealistic.

5. Figures in article are low quality, PF and SF should be marked in Figure 1, Letters in Figures 2 and 4 is blurry, Figure 3b need to mark.

6. In Introduction: Suggest adding reference: hair follicles develop at which stage of rabbit embryo?

Author Response

This study showed that maternal treatment with MT promotes developmental HF in direct offspring, the main mechanisms involved are: the transcriptional regulation of JUN/FOS members of the AP-1 complex emerges as a pivotal factor in mediating the beneficial effects of MT on neonatal hair follicle development. The research is relatively innovative. However, there are still some key questions need to be resolved:

  1. What evidence suggests that MT has crossed the placental barrier? For example, measuring MT in the maternal placenta or fetal blood, or analyzing changes in the maternal uterus internal environments.

Thank you for your precise comments. In fact, there is some evidence indicating that melatonin (MT) can cross the placental barrier in humans and mice [1-3]. Additionally, we have measured the levels of MT in both maternal and fetal serum, and the data have already been published.(Yang F, Gun S. Melatonin supplement induced the hair follicle development in offspring rex rabbits. Journal of Animal Physiology and Animal Nutrition. 2021;105(1):167-74.). The results are presented as follows:

We measured the concentrations of TSH, PRL, and MT in the serum of maternal and fetal rabbits on the day of birth. The results showed that the maternal TSH and PRL levels in the treatment group were significantly lower than those in the control group, while the MT levels were significantly higher in the treatment group than in the control group. Interestingly, PRL was significantly higher in the treatment group of newborn rabbits than it was in the control group, and MT levels were also increased in newborns.

Table 1.Concentration of maternal serum biochemical parameters

Group

TSH(pg/mL)

PRL(pg/mL)

MT(pg/mL)

Con

5.58 ±0.93 a

742.05±156.01 a

32.53±12.00

MT

4.42±1.15

591.58±107.68

46.59±14.26 a

Different lowercase letters in the same column indicate a significant difference(P<0.05), and the same or no letter indicates that the difference was not significant (P>0.05).Con, control group; MT, melatonin; TSH, thyroid stimulating hormone; PRL, prolactin.

Table 2.Concentration of TSH, PRL, and MT in the serum of first-generation rabbits(1-day-old)

Group

TSH(pg/mL)

PRL(pg/mL)

MT(pg/mL)

1-day-old

Con

9.74±1.87

686.68±116.62a

14.27±4.69

MT

9.54±2.75

795.62±52.27b

18.65±5.45

Different lowercase letters in the same column indicate a significant difference (P< 0.05), and the same or no letter indicates that the difference was not significant (P > 0.05). Con, control group; MT, melatonin; TSH, thyroid stimulating hormone; PRL, prolactin.

Also, we have added the references in the manuscript.

  1. Reiter, R.J., et al., Melatonin and stable circadian rhythms optimize maternal, placental and fetal physiology. Human reproduction update, 2014. 20(2): p. 293-307.
  2. Bonde, J.P.E., et al., Risk of miscarriage and occupational activity: a systematic review and meta-analysis regarding shift work, working hours, lifting, standing and physical workload. Scandinavian journal of work, environment & health, 2013. 39(4): p. 325.
  3. Cai, C., et al., The impact of occupational shift work and working hours during pregnancy on health outcomes: a systematic review and meta-analysis. American journal of obstetrics and gynecology, 2019. 221(6): p. 563-576.

  1. MT has a significant impact on the reproductive performance of female rabbits. In this study, Does MT have negative effects such as affecting litter size or live birth? No data was found in the study.

Thank you for your kind suggestion. In our experiment, we observed that melatonin had no significant impact on the breeding performance of female rabbits, and we have already published these findings. (Yang F, Gun S. Melatonin supplement induced the hair follicle development in offspring rex rabbits. Journal of Animal Physiology and Animal Nutrition. 2021;105(1):167-74.). The data are presented as follows:

No female rabbits in the MT group had a miscarriage, nor did any exhibit abnormal behavior or poor maternal behavior. As shown in Table 3,we found that implanting MT during pregnancy had no effect on the reproductive performance of female rabbits.

Table 3.Breeding performance of female rabbits

Group

Litter size

Number born alive

Birth weight/kg

Con

6.56 ±1.42

6.00±1.02

0.34±0.04

MT

7.11±1.83

7.11±1.83

0.38±0.08

The same or no lowercase letter indicates that the difference was not significant (P >0.05).Con, control group; MT, melatonin treatment group.

  1. In the article, relevant genes were identified through skin transcriptome screening, and key genes were tested through RNA and protein analysis. Metabolic profiling was performed on downstream pathways. But they all used relatively single methods, especially in terms of mechanisms, without adding other methods to support them, which feels a bit weak and insufficient for the following conclusions.

We employed transcriptome screening, RNA and protein analysis, immunofluorescence analysis, along with downstream metabolic profiling to infer mechanisms. These methods, which possess good recognition and applicability, were selected as key tools for our initial exploration. Furthermore, we intend to conduct mechanism validation in mice as our next step. We sincerely appreciate your suggestions, as they are crucial feedback for enhancing the quality of our research.

  1. In Methods, At the same day? Sperm of one male rabbit could AI 93 female? This is obviously unrealistic.

In fact, we used frozen semen from the same male rabbit rather than fresh semen. To prevent misunderstanding, we have already made the correction in the original manuscript.

  1. Figures in article are low quality, PF and SF should be marked in Figure 1, Letters in Figures 2 and 4 is blurry, Figure 3b need to mark.

Thank you for the precise comment, we have added them and improved the figures quality.

  1. In Introduction: Suggest adding reference: hair follicles develop at which stage of rabbit embryo?

Thank you for your kindly suggestion. We have add the hair follicle generation in the introductio

Reviewer 2 Report

Comments and Suggestions for Authors

This paper suggests that maternal melatonin (MT) supplementation during pregnancy may enhance fetal hair follicle development by increasing follicle density and reducing apoptosis through the involvement of JUN/FOS and the AP-1 complex expression and MAPK signalling. This may have long-term benefits for fetal hair and skin health. Both the methods and the proof of results are well described, but we would like to point out a few omissions.

1. the study was conducted using a specific animal model (i.e. rabbits), which may not be applicable to human physiology or the complexities of human pregnancy and fetal development. It would be helpful to know why this model was used.

2. Although we have demonstrated several mechanisms of action in this paper, we believe that due to the complexity of biological systems, it will be difficult to isolate the effects of MT as other unmeasured factors may also have a significant impact on hair follicle development.

3. The duration of melatonin administration is not accurately described.

4. Figures 2 and 4 are too low resolution to understand.

Author Response

This paper suggests that maternal melatonin (MT) supplementation during pregnancy may enhance fetal hair follicle development by increasing follicle density and reducing apoptosis through the involvement of JUN/FOS and the AP-1 complex expression and MAPK signalling. This may have long-term benefits for fetal hair and skin health. Both the methods and the proof of results are well described, but we would like to point out a few omissions.

  1. the study was conducted using a specific animal model (i.e. rabbits), which may not be applicable to human physiology or the complexities of human pregnancy and fetal development. It would be helpful to know why this model was used.

Thank you for the recommendation, we have added the reason in the introduction.

  1. Although we have demonstrated several mechanisms of action in this paper, we believe that due to the complexity of biological systems, it will be difficult to isolate the effects of MT as other unmeasured factors may also have a significant impact on hair follicle development.

 We would like to thank the reviewer for the precise recommendation. In the present study, we try to found the mechanisms underlying the function of maternal MT in offspring HF development. Although the other we have taken great care to control for other confounding factors (e.g., by utilizing half-sibling female rabbits and semen derived from the same male Rex rabbit), in order to isolate and elucidate the intrinsic mechanisms associated with the MT treatment.

  1. The duration of melatonin administration is not accurately described.

In the study, the MT microcapsule was implanted once the female rabbits were confirmed to be pregnant, and the capsule was used in vivo from the time of pregnancy until the newborns were born.

  1. Figures 2 and 4 are too low resolution to understand.

Thank you for the precise comment, we have improved the figures quality.

Round 2

Reviewer 1 Report

Comments and Suggestions for Authors

Although we have seen evidence provided by the author to prove that MT could crossed the placental barrier, we still have the following doubts about this article: 

1、 The evidence for the same indicator is single, lacks overall persuasiveness.

2、 The low image quality has not been resolved. 

3、 The metabolome data has not been fully analyzed, and the overall study is relatively simple.

Author Response

Although we have seen evidence provided by the author to prove that MT could crossed the placental barrier, we still have the following doubts about this article: 

Comments 1: The evidence for the same indicator is single, lacks overall persuasiveness.

Response 1:Thank you for your insightful and valuable comments. Indeed, during the design of the experiment, we conducted an extensive review of the relevant literature. Our investigation encompassed the assessment of maternal and offspring serum biochemical parameters, as well as the breeding performance of female rabbits (have been published). Our findings revealed that melatonin (MT) did not exert a significant impact on the reproductive performance of female rabbits but did influence serum levels of thyroid-stimulating hormone, prolactin (PRL), and melatonin itself (Figure1a,1b). To ascertain whether MT adversely affected lactation, we monitored the offspring rabbits until they reached five months of age. No discernible differences were observed between the MT-treated and control offspring in terms of pre-weaning mortality, disease incidence, or behavior. As illustrated in Figure 1.c, there was no statistically significant difference in the growth performance between the two groups of offspring rabbits. Furthermore, the influence of maternal MT on PRL levels in the serum of young rabbits persisted until the neonatal rabbits attained adulthood. Consequently, the administration of low-dose MT to pregnant rabbits appears to be safe. Additional, neonatal rabbits in the MT group exhibited higher MT levels compared to the control group, whereas rabbits at five months of age displayed lower MT levels. The serum PRL concentration in the MT group was significantly greater than that in the control group for both newborn and five-month-old rabbits (P < 0.05) (Figure 2.a, b). These results indicate that maternal MT implantation affects the PRL concentration in offspring, which may provide an explanation for the observed impact on the development of their hair follicles (HFs). These serum hormone changes could provide evidence that maternal melatonin (MT) crosses the placenta and subsequently affects the fetus. These data have been published, and we have been referencing it in the manuscript.

Figure 1 :(a): Reproductive performance of female rabbits with and without MT; (b):The concentrations of maternal serum biochemical parameters in rabbits with and without MT implantation; (c): Body weight and body length of 5-months-old rabbits.

No p values label means no significance (p > 0.05).

Figure 2 The concentrations of TSH, PRL, and MT in the serum of first-generation rabbits

(a): 1-day old; (b): 5 months old

No p values label means no significance (p > 0.05).

In addition to this, numerous studies have demonstrated MT could cross the placenta in various species. The pineal gland attains maturity postnatally, with fetuses and neonates being incapable of producing melatonin (MT). Instead, mammalian embryonic development relies on maternal MT supplied via the placenta, which traverses the placental barrier without undergoing modification[1]. In rodents, MT binding sites have been identified in the fetal pituitary gland, and maternal blood MT concentrations escalate with gestational progression, peaking at the time of parturition [2, 3]. Maternal MT implantation can permeate the placenta to enter fetal circulation, thereby acting as a crucial regulator of circadian rhythms in the fetus. It directly contributes to the development of the nervous and endocrine systems and safeguards embryonic organs from oxidative stress damage during development [4]. In humans, exposure to night work during pregnancy can disrupt maternal MT secretion, disturbing the internal circadian rhythm and potentially elevating the risks of preterm birth and low birth weight [5, 6]. Notably, following a regimen of daily MT injections aimed at normalizing circadian rhythm disorders, fetal clock gene expression reverts to normalcy [7]. Additionally, MT administration during pregnancy has been shown to mitigate arsenic-induced damage in rat offspring, inhibiting inflammation, DNA damage, and apoptosis in these offspring  [8]. Furthermore, it reduces DNA damage in hepatocytes and blood cells of newborn mouse brains [9], safeguarding against programmed cardiovascular maladies, and emerges as a potential therapeutic target for clinical intervention targeting developmental origins of health and disease [10]. Collectively, these studies underscore the capacity of MT to traverse the placental barrier from mother to offspring, inducing alterations in the uterine microenvironment and fetal reprogramming. These alterations can exert profound effects on the organism throughout its entire life cycle. Our findings indicate that maternal melatonin (MT) can enhance hair follicle density and mitigate hair follicle cell apoptosis. We hypothesize that these effects are mediated through the placental transfer of MT. Specifically, our results suggest that the maternal administration of MT positively influences hair follicle health in the offspring, potentially via the placental route, thereby promoting an increase in follicle density and a reduction in cellular apoptosis within the hair follicles.

Additional, to identify the regulatory factors involved, we conducted a combined analysis using transcriptomics and metabolomics. During the analysis of transcriptome data, we conducted an extensive review of relevant literature. FOS and JUN are key genes screened out through transcriptome sequencing based on gene significance score, bioinformatics analysis, correlation analysis with primary and secondary hair follicles, gene interaction analysis, and metabolite interaction analysis. We have confirmed the differential expression of these two genes and proteins between the treatment group and the control group using qPCR, WB protein detection, and immunohistochemical detection. FOS and JUN play an important role in hair follicles. FOS occupies a central position in the regulation of cellular processes such as proliferation, differentiation, transformation, and apoptosis. JUN constitutes a vital member of the JUN family of proteins and possesses the capacity to form homodimers with proteins belonging to the FOS family, resulting in the formation of the dimer activator protein-1 (AP-1). This protein complex is implicated in a wide array of cellular activities, encompassing cell proliferation, differentiation, and apoptosis. Available evidence underscores the pivotal role of the AP-1 transcription factor in the development and growth of mammalian embryos. Therefore, we hypothesize that the impact of maternal melatonin (MT) on offspring hair follicles is mediated through the regulation of FOS/JUN during the embryonic period. Changes in AP-1 levels in the serum of offspring rex rabbits can also corroborate our hypothesis.

Next, we aim to corroborate our research findings at the cellular level, thereby delineating our ongoing and prospective research trajectory. We are deeply appreciative of your suggestions, which have significantly bolstered our confidence in pursuing further inquiries into the effects of MT on offspring hair follicles. In the subsequent phase of our investigation, we anticipate focusing specifically on hair follicle cells that are impacted by MT, thereby presenting a unique and distinct research pathway.

Certainly, we have incorporated additional results and an enhanced discussion into the manuscript, adhering to the reviewer's suggestions.

  1. Sagrillo-Fagundes, L., et al., Melatonin in pregnancy: Effects on brain development and CNS programming disorders. Current pharmaceutical design, 2016. 22(8): p. 978-986.
  2. Serón-Ferré, M., et al., Circadian rhythms in the fetus. Molecular and cellular endocrinology, 2012. 349(1): p. 68-75.
  3. de Almeida Chuffa, L.G., et al., Melatonin promotes uterine and placental health: potential molecular mechanisms. International journal of molecular sciences, 2020. 21(1): p. 300.
  4. Voiculescu, S., et al., Role of melatonin in embryo fetal development. Journal of medicine and life, 2014. 7(4): p. 488.
  5. Bonde, J.P.E., et al., Risk of miscarriage and occupational activity: a systematic review and meta-analysis regarding shift work, working hours, lifting, standing and physical workload. Scandinavian journal of work, environment & health, 2013. 39(4): p. 325.
  6. Cai, C., et al., The impact of occupational shift work and working hours during pregnancy on health outcomes: a systematic review and meta-analysis. American journal of obstetrics and gynecology, 2019. 221(6): p. 563-576.
  7. Mendez, N., et al., Timed maternal melatonin treatment reverses circadian disruption of the fetal adrenal clock imposed by exposure to constant light. PLoS One, 2012. 7(8): p. e42713.
  8. Abdollahzade, N., S. Babri, and M. Majidinia, Attenuation of chronic arsenic neurotoxicity via melatonin in male offspring of maternal rats exposed to arsenic during conception: Involvement of oxidative DNA damage and inflammatory signaling cascades. Life Sciences, 2021. 266: p. 118876.
  9. de Sousa Coelho, I.D.D., et al., Protective effect of exogenous melatonin in rats and their offspring on the genotoxic response induced by the chronic consumption of alcohol during pregnancy. Mutation Research/Genetic Toxicology and Environmental Mutagenesis, 2018. 832: p. 52-60.
  10. Hansell, J.A., et al., Maternal melatonin: Effective intervention against developmental programming of cardiovascular dysfunction in adult offspring of complicated pregnancy. Journal of Pineal Research, 2021. 72(1): p. e12766.   

    Comments 2: The low image quality has not been resolved. 

    Thank you for you kindly comment, we have redone the figures and made them into high quality vector graphics for easier viewing.

    Comments 3: The metabolome data has not been fully analyzed, and the overall study is relatively simple.

    We have conducted a comprehensive analysis of the metabolome data, with a specific emphasis on two key metabolites. However, owing to limitations in space, our main focus was on discussing the downstream metabolites of the selected genes, FOS and JUN, which led to a concise presentation of the metabolome analysis. We are profoundly grateful for the reviewer's suggestions and have accordingly incorporated the analysis results in the supplementary file that we have uploaded to address this section. The supplementary information encompasses: the detailed screening process of the metabolome data, the intricate interactions among the metabolites, the pathways enriched by these metabolites, and an integrated pathway analysis of both the metabolome and transcriptome. These additions are intended to enhance the comprehension of the content presented in this paper for other researchers.

    Special gratitude is extended to you for your kind and valuable comments. We have endeavored to enhance the manuscript and address all feedback meticulously. We sincerely appreciate the diligent efforts of the Editors and Reviewers and hope that our revisions meet with your approval. Once again, thank you immensely for your insightful comments and suggestions.
